# On Metacognition: Overconfidence in Word Recall Prediction and Its Association with Psychotic Symptoms in Patients with Schizophrenia

**DOI:** 10.3390/brainsci14090872

**Published:** 2024-08-29

**Authors:** Yvonne Flores-Medina, Regina Ávila Bretherton, Jesús Ramírez-Bermudez, Ricardo Saracco-Alvarez, Monica Flores-Ramos

**Affiliations:** 1Subdirección de Investigaciones Clínicas, Instituto Nacional de Psiquiatría, Mexico City 14370, Mexico; saracco@inprf.gob.mx; 2Dirección de Servicios Clínicos, Instituto Nacional de Psiquiatría, Mexico City 14370, Mexico; regisbre.14@gmail.com; 3Unidad de Neuropsiquiatría, Instituto Nacional de Neurología y Neurocirugía, Mexico City 14370, Mexico; jesusramirezb@yahoo.com.mx; 4Laboratorio de Epidemiología Clínica, Subdirección de Investigaciones Clínicas, Instituto Nacional De Psiquiatría, Mexico City 14370, Mexico; flores_ramos@hotmail.com

**Keywords:** schizophrenia, metacognition, metamemory, overconfidence, psychotic symptoms

## Abstract

A two-factor account has been proposed as an explanatory model for the formation and maintenance of delusions. The first factor refers to a neurocognitive process leading to a significant change in subjective experience; the second factor has been regarded as a failure in hypothesis evaluation characterized by an impairment in metacognitive ability. This study was focused on the assessment of metacognition in patients with schizophrenia. The aims of the study were to measure the overconfidence in metacognitive judgments through the prediction of word list recall and to analyze the correlation between basic neurocognition (memory and executive function) and metacognition through a metamemory test and the severity of psychotic symptoms. Method: Fifty-one participants with a diagnosis of schizophrenia were evaluated. The Positive and Negative Syndrome Scale (PANSS) was used to assess the severity of psychiatric symptoms, and the subtest of metamemory included in the Executive Functions and Frontal Lobe-2 battery (BANFE-2) was used to evaluate overconfidence and underestimation errors, intrusion and perseverative response, total volume of recall, and Brief Functioning Assessment Scale (FAST) for social functioning. Results: The strongest correlation is observed between overconfidence errors and the positive factor of the PANSS (r = 0.774, *p* < 0.001). For the enter model in the multiple linear regression (r = 0.78, r^2^ = 0.61; F = 24.57, *p* < 0.001), the only significant predictor was overconfidence errors. Conclusion: Our results highlight the relevance of a metacognitive bias of overconfidence, strongly correlated with psychotic symptoms, and support the hypothesis that metacognitive defects contribute to the failure to reject contradictory evidence. From our perspective, these findings align with current mechanistic models of schizophrenia that focus on the role of the prefrontal cortex.

## 1. Introduction

The scientific explanation of psychosis remains a challenge, with significant implications for clinical practice in the field of psychopathology. Several frameworks have been developed to explain this process. Mechanistic models accounting for abnormalities in brain structure and signaling are necessary to understand psychosis; for instance, molecular imaging research on patients with a schizophrenia diagnosis shows an increase in dopamine synthesis in the striatum [1]. However, cognitive frameworks have been regarded as a necessary mediation between the level of dysfunctional mechanisms and the level of clinical phenomenology [2]. The explanation of the clinical features of psychosis in schizophrenia requires a comprehensive approach not only to symptoms such as delusions and hallucinations, but also to the psychopathology of self-awareness in schizophrenia [3,4]. In this article, we will focus on psychotic features as these are measured by the PANSS instrument, with a particular emphasis on delusions, as these phenomena have been widely discussed by the tradition of cognitive neuroscience. For instance, in the cognitive framework, a two-factor account has been proposed as an explanatory model for the formation and maintenance of delusions in patients across the spectrum of psychosis [5,6,7]. The first factor refers to a neurocognitive process leading to a significant change in subjective experience that is felt as unexpected, salient, or abnormal by the individual [7,8]. The second factor has been regarded as a failure in hypothesis evaluation. It considers the process from unexpected observations to the formation of new beliefs. More specifically, it has been characterized as a failure to reject hypotheses in the face of disconfirmatory evidence [5,7]. During everyday interactions, individuals often make incorrect inferences about their environment, others, or themselves. However, metacognitive abilities allow them to correct these errors when presented with disconfirming evidence or through critical reasoning. In the state of psychosis, however, false ideas or perceptions can be maintained tenaciously despite contradictory evidence. Some researchers in the field of cognitive science postulate that this second factor is an explicit metacognitive dysfunction, or a failure in the implicit monitoring and control of cognitive accuracy, which has been regarded by some theorists as implicit metacognition [5,9].

Although the two-factor model has been developed for the analysis of delusions, it is helpful as well to understand the evaluation of hallucinations by the individual and, thus, the disturbance in perceptual reality monitoring that is frequent in patients suffering from psychotic states [10]. In order to make these two factors explicit, we will describe them in the field of psychosis related to patients with schizophrenia. The first hit, or the neurocognitive mechanisms, in the model described above has been classically described as a problem in the neuropsychological mechanisms; the perception was the initial cognitive domain studied in cases of delusional misidentification [11]. However, in the attempt to make an explanation of psychosis in schizophrenia with the resources of a two-factor account, we must consider that the neuropsychological impairments observed in this population involve several cognitive domains, including perception, but also processing speed, attention, memory, executive function, and emotional processing [12,13]. These deficits are observed in ultra-high-risk individuals, chronic and late-onset schizophrenia-like patients [13,14,15,16]. Cognitive performance is heterogeneous, ranging from mild impairment to dementia-like syndromes, and it is significantly related to psychosocial functionality and insight [12,13]. The cognitive deficits have been observed consistently in patients with first-episode psychosis before the use of antipsychotic medication [17]. However, this basic neurocognitive factor is not sufficient for the generation of psychotic states in schizophrenia [5,7], but the presence of this kind of information processing can also be seen in nonclinical subjects who have had psychotic experiences, according to community samples [18].

The second hit can be characterized as an impairment of implicit or explicit metacognitive ability [5,7]. The term metacognition refers to the processes that enable us to evaluate our own mental operations through monitoring, self-regulation, and control of cognitive mechanisms. This leads to the capacity to recognize accurately, reflect, and regulate our own mental representations [19]. The metacognitive skills allow the person to identify a particular mental state, for example, experiencing a particular visual or auditory perception, executing a motor act, or accurately experiencing the evocation of information. This has been called metacognitive awareness [1,19,20]. Additionally, these skills allow us to make judgments about these phenomena and compare incoming information to modify or rectify the judgment. This is called metacognitive confidence [16,18,19,21]. Finally, there is a predictive aspect to metacognition that enables us to proofread our own cognitive performance [18]. Metacognitive dysfunction has been associated with poor outcomes in educational, work, and social behavior. At the clinical level, a significant association between metacognitive deficits and the maintenance of delusional beliefs has been shown [22]. For research and practical purposes, it is important to distinguish between metacognition and other related concepts, such as anosognosia, insight, and theory of mind. We regard anosognosia as the unawareness of a functional deficit, following focal or diffuse brain damage [23,24], whereas the term insight refers to a type of self-knowledge, “which individuals hold not only about the disorder affecting them but also about how the disorder affects their interaction with the world.” [25]. Theory of mind, on the other hand, can be defined as the use “of folk psychological knowledge and heuristics to think about one’s own and other people’s mental states” [26]. In this article, we will focus on the concept of metacognition.

Several resources have been developed to assess metacognitive deficits in patients with schizophrenia, including the Indiana Psychiatric Illness Interview, the Metacognitive Assessment Interview, the Metacognitive Self-assessment Scale, the Metacognition Assessment Scale-abbreviated, the Metacognition Questionnaire-30, as well as memory paradigms. For instance, the feeling of knowing an instrument or the metamemory inventory in adulthood. The comparisons between groups of healthy subjects and patients with schizophrenia predominantly show a lower performance of the patient across the different measurement methods [22,23,24,25,26,27,28]. It is also possible to observe a significant correlation between the scores on these instruments and the degrees of severity in several symptomatologic domains and functional outcomes [28,29,30].

One of the limitations of metacognition assessment in clinical settings is the length of time required for the assessment. Additionally, the procedures that rely on exploring the patients’ beliefs without measuring cognitive performance data are questionable in terms of validity and objectivity. A way forward is to measure the overconfidence in judgment through the assessment of the correspondence between the beliefs about performance and the objective cognitive performance. A novel approach to existing paradigms for the assessment of metacognitive judgment in patients with schizophrenia is the use of a simple and quick metamemory evaluation, a word list learning task. In this procedure, participants memorize a list of words. Before each recall trial, they are asked to predict the number of words they will be able to recall. Throughout the trials and presentations of the same word list, subjects have the opportunity to receive feedback on their performance and narrow the gap between their predictions and actual performance. The results depend on the accuracy of the relationship between the prediction and the recall. The number of words above or under the prediction can be subtracted from the number of words evoked. In patients with frontal damage, Luria [31] and Vilkki [32] described this methodology for the assessments of metacognition, and more specifically, metamemory.

This study was focused on the assessment of metacognition in patients with schizophrenia and describing a new approach to measuring the metamemory phenomenon in a clinical setting with a word-list learning task. The aims of the study were: (1) to measure the overconfidence in metacognitive judgments through the prediction of word list recall; (2) to analyze the correlation between basic neurocognition (memory and executive function), metacognition (through a metamemory test), and the severity of psychotic symptoms. We expect a positive correlation between errors in recall prediction and the severity of the positive symptom factor of the PANSS; also, we expect that overconfidence explains a higher variance in positive symptoms through a prediction model. (3) Finally, we expected to find a correlation between overconfidence scores and levels of functionality.

## 2. Materials and Methods

### 2.1. Participants and Setting

The study was approved by the Ethics and Research Committee of the Instituto Nacional de Psiquiatría Ramón de la Fuente Muñiz (INPRFM), National Institute of Psychiatry of Mexico, with number of the Comité de etica institucional (CEI): CEI/C/029/2022.

Participants and selection criteria. A total of 51 participants were included. The participants met the following inclusion criteria: (a) diagnosis of schizophrenia, (b) age of 18 years or more, (c) minimum schooling of six years, (d) currently receiving antipsychotic treatment, and (e) at least one year with the diagnosis. We excluded participants with delusional disorder, substance-induced psychosis, schizoaffective disorder, major neurocognitive disorder or any other serious psychiatric comorbidity, catatonic syndrome, or active consumption of any substance other than tobacco. Severe or uncontrolled medical illnesses were also discarded. Additionally, we used strict criteria to exclude patients with a reasonable suspicion of suffering from an underlying neurological or systemic cause for their psychotic state, including endocrine disease, nutritional pathologies, autoimmune pathologies. We also excluded patients with “late onset schizophrenia” and “very late onset schizophrenia”. All participants signed an informed consent letter.

### 2.2. Procedure and Data Collection

We conducted an observational study, in a cross-sectional evaluation [33]. Convenience sampling was used to recruit participants. The clinical diagnosis of schizophrenia was made by a specialized psychiatrist, following the Diagnostic and Statistical Manual of Mental Disorders, 5th version (DSM-5) and medical records. After verifying inclusion/exclusion criteria, participants signed an informed consent letter. The researcher’s assistants with medical degrees were trained in the administration of the tests. Under the supervision of the principal investigator, the assessment was performed in a single session, in a quiet, sound-controlled, and illuminated room. The assessment was conducted in the following order: registration of demographic and clinical data, PANSS interview, metacognitive test, and FAST. The complete evaluation lasted approximately 90 min.

Demographic and clinical data were obtained with a structured interview: age, years of education, age of onset, illness duration, hospitalizations, and pharmacological treatment.

Metacognition test. The subtest of metamemory included in the Executive Functions and Frontal Lobe Battery (BANFE-2) was used [34]. The test consists of learning a nine-word list presented in the same order over 5 trials (pear, tube, cow, boat, eraser, sandpaper, hand, bow, and letter). Prior to each trial, participants must indicate how many words they believe they will remember, and this number is recorded. Subsequently, the list of words is read aloud, and the participant is asked to recall as many words as possible. The number of words recalled is then recorded. The difference between the predicted number of words and the actual number recalled is calculated. A positive difference (where the prediction exceeds the actual recall) is considered an indication of overconfidence or overestimation, while a negative difference (where the prediction falls short of the actual recall) is considered an underestimation. A score of zero indicates equivalence between prediction and performance. This assessment battery has been adapted and standardized for the Mexican population and is sensitive to academic years or school experience. The normative data adjusted for age and schooling were also recorded for the overconfidence and underestimation errors to determine the level of impairment. Memory assessment. Total recall over 3 trials of word list was obtained. Each trial is scored with one point for each word recalled. The total recall is obtained by averaging the number of correct responses reported in the three trials. The normative data adjusted for age and schooling for the performances were obtained to determine the level of impairment. Perseverative and intrusive words expressed by the participants were recorded during the 5 trials in the subtest of metamemory. These responses are considered proxy indicators of executive dysfunction. The first were classified within the domain of inhibitory control, and the second were classified within the domain of interference inhibition.

Psychopathology. The Positive and Negative Syndrome Scale (PANSS) was used to assess the severity of psychiatric symptoms. In the present study, the adaptation for Mexican population was used. Five main factors have been identified positive, negative, cognitive, anxiety/depression, and excitation [35]. The factor of positive symptoms, including delusions and hallucinatory behavior, is used as a measure of psychotic states.

Social Functioning. We used the Brief Functioning Assessment Scale (FAST) to evaluate the following aspects: autonomy, work activity, cognitive functioning, finances, interpersonal relationships, and free time. This instrument has been adapted into a Spanish version for its use in patients with schizophrenia [36].

### 2.3. Statistical Analysis

Descriptive statistics were performed for clinical and sociodemographic data, using media and proportions as applicable. The Pearson correlation test was used to assess the relationships between positive and negative factors of the PANSS and FAST scores with the overconfidence errors, underestimation errors, total recall of memory assessment, and the number of perseverative and intrusive responses. We used Bonferroni for multiple comparisons to correct the correlations. An additional correlation analysis was carried out with the variable’s positive symptoms, overconfidence errors, underestimation errors, and total memory recall with age, years of education, age of onset, and illness duration to identify cofounders. A multiple linear regression was conducted with variables significantly related to the positive symptoms. Analysis was performed using the program JASP version 18.0; the significance level was set at 0.05.

## 3. Results

A total of 51 participants were evaluated: 22 women and 29 men. The demographic and clinical data are shown in Table 1. In our sample, patients exhibited moderate severity of symptoms according to the PANSS, despite pharmacological treatment (Aripiprazole 18%, Risperidone 18%, Haloperidol 14%, Olanzapine 14%, Clozapine 10%, Sulpiride 8%, Paliperidone 8%, Perfenazine 4%, Quetiapine 4%, and Trifluoperazine 2%). These data indicate the presence of both positive and negative symptoms, which are evident in the sample. Additionally, the assessment of memory indicates that mild, moderate, and severe verbal memory impairments are identifiable in 92.1% of the sample. Regarding metacognitive performance, overconfidence errors were present in 51% of the patients compared to normative data, while a lower percentage of 27.4% showed underestimation errors compared to normative data (Table 2).

Initial analysis (before Bonferroni correction) revealed significant correlations of positive symptoms with overconfidence errors (r = 0.794 ***), total errors of the metamemory test (r = 0.417 **) and the intrusion response (r = 0.356, *p* = 0.01), and FAST (r = 0.619, *p* = 0.01). In Figure 1, we show the correlation coefficients (Pearson’s test) between the psychopathological scores, the social functioning test, and the scores on the metamemory test after the Bonferroni test for multiple comparisons. As we expected, the strongest correlation is observed between overconfidence errors and the positive factor of the PANSS (r = 0.774, *p* < 0.001). Unexpectedly, we found significant correlations between overconfidence errors and the excitation factor, as well as with the PANSS and the FAST total scores. The correlations between PANNS positive scores and overconfidence errors with sociodemographic and clinical variables such as age (r = 0.18, r = 0.06), years of education (r = −0.08, r = −0.10), age of onset (r = −0.07, r = −0.08), or disorder duration (r = 0.21, r = 0.10) were not significant. No significant correlations were observed between memory performance and overestimation (r = −0.34) or underestimation errors either (r = 0.26).

The multiple linear regression model for predicting positive symptoms was conducted with variables significantly related to the positive symptoms before Bonferroni correction: overconfidence errors (r = 0.794, *p* = 0.0001), total errors of metamemory test (r = 0.417, *p* = 0.001), and the intrusion response (r = 0.356, *p* = 0.001). For the entry model, the results were r = 0.78, r^2^ = 0.61; F = 24.57 and *p* < 0.001. The only significant predictor in the model was overconfidence errors. Table 3 shows the model summary, and Figure 2 shows the partial regression plots.

## 4. Discussion

The background problem that gives rise to this study regards the explanation of psychosis through the study of metacognition in patients with a diagnosis of schizophrenia. The study of metacognitive dysfunction contributes to the explanation of the psychotic states, and more specifically, to the onset and maintenance of delusions. To approach this general question from an empirical perspective, we studied the overconfidence in metacognitive judgments through the prediction of a word list recall in a sample of patients with schizophrenia. As the main result, we observed an increased number of overconfidence errors, which implies that patients tend to overestimate their own ability to remember a list of words. This finding is consistent with previous descriptions of patients with schizophrenia regarding the presence of overconfidence judgments about memory capacities. Although our study focused on verbal memory, it is interesting to observe that the failures in the metamemory process have been identified in verbal, visual, and episodic forms of memory by means of different stimulus presentations: word pairs, word lists, static images, and videos [37,38,39,40,41].

We used the two-factor model of delusions as an explanatory framework to approach the problem of psychosis in our sample, as this model is compatible with the study of implicit or explicit metacognition. In our own adaptation of the model, factor 1 was approached through basic neurocognitive measures of verbal memory, and with the measurement of intrusive and perseverative responses, as proxies for the executive control of memory retrieval. We did not find a correlation between positive symptoms and perseverative responses or total memory recall. However, we observed a correlation between the positive symptom factor of the PANNS and the number of intrusion errors before the Bonferroni test, which can be regarded as a form of interference due to failures in inhibitory control. This proxy for the assessment of executive functioning points to the possible influence of these variables on the generation or maintenance of the positive symptoms of psychosis. As is known, the exact relationship between executive functioning and psychosis remains elusive [41]. It has been described that failures in several executive processes are associated with the presence of psychopathology in general terms, but limited evidence is available on their specific association with psychosis [42,43]. White et al. [44] showed significant associations between psychotic symptoms and working memory, response inhibition, attention vigilance, and cognitive flexibility. Interestingly, these authors did not observe correlations with impulsivity, which is consistent with our study. This finding and the available literature lead us, along with other authors, to believe that the nature of the first hit is mostly heterogeneous. In their most recent review, Lee et al. [17] show the significant variability in the scores observed in patients with psychosis; this heterogeneous and variable profile is observed not only in executive functions, but also in processing speed, attention, memory, and learning. Although not enough literature is available in this regard, we do not reject the possibility that other executive function impairments that have been widely reported in patients with psychosis, such as failures in working memory, could have a greater weight as basic neurocognitive disturbances contributing to the generation of positive symptomology.

In our study, however, the logistic regression analysis showed that only metacognitive errors predicted psychotic symptomatology. This is consistent with the proposal by Coltheart and Davies [5,7] who state that the first factor is not sufficient for the generation of delusions, and that a bias against disconfirmatory evidence (BADE) is necessary to explain the maintenance of delusions. This approach has also been replicated in other models with different metacognitive measures [6,37,40,44,45].

An alternative formulation to BADE has been proposed by one-factor theorists based on a predictive processing account. According to this explanatory framework, delusions are conceptualized as beliefs that evolve over time, involving a negotiation between existing beliefs and the available evidence under conditions of uncertainty. In this framework, unexpected experiences may trigger the formation of delusional beliefs. However, once this belief is formed, it “creates new expectations about uncertainty that reduce updating but also facilitate the flexible assimilation of contradictory evidence” [46]. From our perspective, the predictive processing framework is useful as a theoretical tool for the explanation of psychosis; however, our results highlight the relevance of a metacognitive bias of overconfidence, strongly correlated with psychotic symptoms. Thus, our study supports the hypothesis of metacognitive defects contributing to the failure to reject contradictory evidence.

Additionally, the role of basic functions, such as attention, should be considered in the metacognitive phenomenon; however, this topic remains in intense debate. Matthews and colleagues [47] explored the codependency between top-down attention in the performance of the visual discrimination task and confidence rating to calculate metacognition. They showed that in a dual task, in which individuals must simultaneously attend to two tasks and two types of stimuli, confidence in perceptual judgments remains identical on correct and incorrect trials (presence and absence of attention) but showed a significant correlation between correct trials and attention. Likewise, the authors indicate that the metacognitive accuracies are much higher on tasks of lower complexity and decrease with increasing cognitive demand. However, Rech, Mamassian, and Gardelle [48] show in their work that confidence judgments are sensitive to involuntary changes in attention; that is, individuals may show greater confidence in their assertions when cues are presented and facilitate the attentional process. To solve this dilemma in our task, we require that we perform a specific measure of attention in our patients, this is considered a limitation of this study, but also a hypothesis to explore in this type of measure.

The results of our study highlight the relationship between metacognition and psychotic symptoms in a sample of patients with schizophrenia. From our perspective, these findings align with current mechanistic models of schizophrenia that focus on the role of the prefrontal cortex. Metacognitive abilities related to memory tasks have been found to be impaired in patients with frontal lobe damage; for instance, lesions affecting Brodmann areas 10 and 46 in humans and monkeys lead to changes in confidence formation, but not first-order task performance. The connectivity of the prefrontal cortex with other cortical and subcortical structures is relevant to understanding the development of metacognition [19]. As known, prefrontal abnormalities in structure and function are among the most consistently observed in patients with schizophrenia: for instance, disturbances in cortical thickness of the lateral and medial aspects of the prefrontal cortex, with a moderate size effect (Cohen’s *d*), as compared to healthy subjects matched for demographic variables [49].

Our results showed an unexpected correlation between overestimation scores and excitability symptoms: hostility, uncooperativeness, and poor impulse control. In this regard, literature has postulated that metacognitive skills are not directly related to the presence of aggressive or hostile behavior; however, these skills play a mediating role in the occurrence of hostile behavior when individuals show poor performance in monitoring abilities and simultaneously present a significant reactivity to emotions such as anger [50]. The interplay between metacognition and impulsivity is challenging because we are not aware of an integrative framework providing a full explanation of the connection between these variables. However, different theories have been proposed: the conflict monitoring theory, the integrative self-control theory, the process model of self-control, the metamotivation model, the trait models, and the self-regulated learning and problem-solving models [51]. Marie Hennecke and Sebastian Bürgler [51] propose that there are two sets of metacognitive components, including self-awareness and metacognitive regulatory processing, that function during and after self-control behavior. According to this perspective, failures in either component could potentially influence the presentation of poor impulse control behaviors. Thus, our findings could indicate that overconfidence errors are related to the presence of hostility and impulse control failures in our patients, since overconfidence is indicative of a monitoring problem or a difficulty in self-awareness.

Finally, we found a correlation between overconfidence scores and levels of social functioning. It has been postulated that metacognitive skills are the bridge between neurocognitive skills and real-life outcomes, such as work, school, and the generation of interpersonal relationships. Specifically, it has been proposed that having an accurate perception of our abilities and limitations allows us to more effectively guide our behavior [50]. In patients with schizophrenia, it has been observed that overestimation predicts impaired outcomes in residential, social, and vocational domains [52,53,54]. In this sense, our findings point in the same direction, showing that levels of overconfidence are related to worse performance on the total score of a functionality test that includes social, occupational, hobbies, and financial domains. Metacognition has also been shown to play a moderating role in other complex phenomena, such as clinical insight or the tendency to be unaware of what others perceive as changes or alterations. Lysaker and Chernov [55] showed the first model of overlap between clinical insight, cognitive insight, and metacognition. Their results indicate that self-reflection and clinical insight were significantly mediated by metacognition and positive symptoms. An interesting line to develop is the relationship between the measure proposed in the present study and clinical insight.

The strengths of this study rely on the utility of a friendly measure for the clinical study of metacognition in patients with chronic psychosis. However, our study has important limitations: the sample size is discrete, which does not allow generalizations, several variables, including body mass index, chlorpromazine equivalents, duration of untreated psychosis, global clinical impression, personal and social performance, and global assessment of functioning, may be of interest for further analysis; on the other hand, our clinical population is limited to patients with schizophrenia. Thus, we do not know if this same phenomenon can be replicated with other psychotic disorders. A broader evaluation of the different neurocognitive and metacognitive domains could be helpful in the future to develop more specific proposals on a two-strike theory of psychosis.

In conclusion, overconfidence, understood as the tendency to have excessive confidence in one’s own beliefs and judgments, is related to the presence of positive symptoms in patients with schizophrenia. This finding suggests that in the treatment of patients with schizophrenia, the evaluation of this dimension of metacognition could allow us to explore other interventions that could help reduce positive symptoms.

## Figures and Tables

**Figure 1 brainsci-14-00872-f001:**
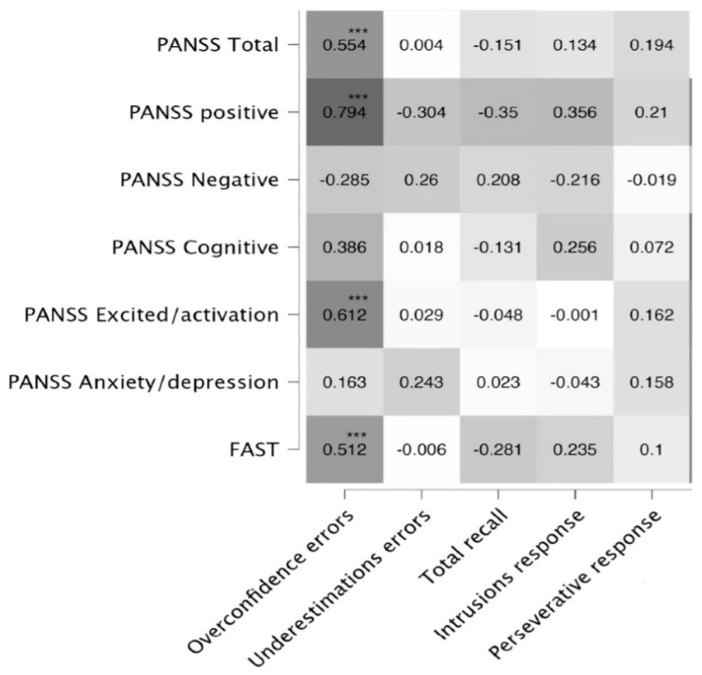
Heatmap of Pearson’s correlations between overconfidence errors, underestimation errors, total recall in memory, intrusion, and perseverative responses; and PANSS total score, as well as the five-factor scores and the FAST score. *** *p* < 0.001. After Bonferroni’s correction.

**Figure 2 brainsci-14-00872-f002:**
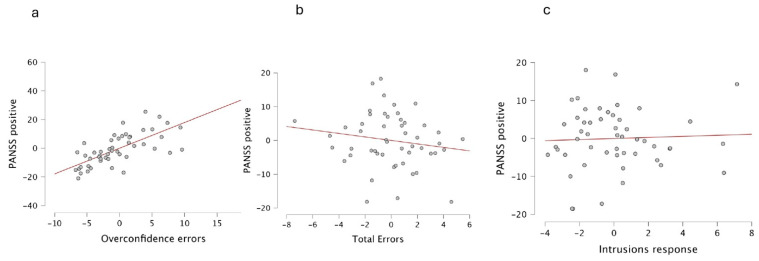
Regression Plots. The model was constructed using positive symptoms as dependent variable, and overconfidence errors, total or metacognitive errors, and intrusion responses as predictors. We observed a strong correlation between overestimation errors (panel **a**) and positive symptoms but not with the total or metacognitive errors (panel **b**) or the intrusion response (panel **c**).

**Table 1 brainsci-14-00872-t001:** Demographic and clinical characteristics of the sample.

Demographics. N = 51	Mean	SD	Minimum	Maximum
Age (years)	42.7	11.6	18	67
Years of education	11.9	2.6	6	16
Age of onset (years)	25.1	7.5	15	47
Disorder duration (years)	17.6	10.8	1	45
Hospitalizations	1.5	1.8	0	7
Psychopathology and Functionality				
PANSS total	79.6	26.7	36	150
PANSS positive	25.9	14.6	8	53
PANSS negative	17.7	10.9	7	43
PANSS Cognitive	22.6	8.2	8	41
PANSS Anxiety/depression	8.4	4.5	4	25
PANSS Excited/activation	7.2	5	4	25
FAST	41.9	18.46	6	72

SD: standard deviation; PANSS: the Positive and Negative Syndrome Scale. FAST: the Brief Functioning Assessment Scale.

**Table 2 brainsci-14-00872-t002:** Metamemory, Memory, and Executive Function Performance.

	Mean	SD	Minimum	Maximum
Overconfidence errors	6	5.7	0	21
Normative scores	7	5	1	14
Percentage of patients in the impairment classifications.	Normal 49%	Mild 2%	Moderate 13.7%	Severe 35.3%
Underestimation errors	2.8	3.6	0	13
Normative scores	9	4	1	14
Percentage of patients in the impairment classifications.	Normal 72.5%	Mild 5.9%	Moderate 3.9%	Severe 17.6%
Metamemory total errors	10.1	4.12	2	21
Total recall, memory assessment	3	1	1	6
Normative scores	4	2	1	10
Percentage of patients in the impairment classifications.	Normal 7.8%	Mild 13.7%	Moderate 33.3%	Severe 45.1%
Intrusions response	2.3	3	0	11
Perseverative response	2.3	3.3	0	21

Normative scores are obtained by comparing the participant’s performance with his or her normative group without pathology. The normative scores are scaled with a range of 1–19; 6–4 points imply mild to moderate impairment and 3-1 severe impairment [34].

**Table 3 brainsci-14-00872-t003:** Summary of the multiple linear regression model for predicting positive symptoms in patients with schizophrenia.

Coefficients	Unstandardized	Standardized	t	*p*
Positive errors	1.82	0.77	8.46	<0.001
Total errors	−0.49	−0.17	−1.08	0.28
Intrusion response	0.44	0.09	0.92	0.36

## Data Availability

Data is unavailable due to privacy.

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
