# Peer review of "On Metacognition: Overconfidence in Word Recall Prediction and Its Association with Psychotic Symptoms in Patients with Schizophrenia"

_brainsci, 2024, doi:10.3390/brainsci14090872_

Round 1

Reviewer 1 Report

Comments and Suggestions for Authors

Please see the attachment for detailed revision comments

Author Response

  1. We are very grateful to our reviewer, the comments helped us to improve our text and below are the answers to his comments.

    1. Please provide a institutional email.

    Response: We add: [email protected]

    1. Explain all Acronyms. Please include FAST tool in your abstract. Explain if the Author used BANFE or BANFE-2; no capitalize the first letter for schizophrenia nor metacognition

    Response: We add all acronyms line 31-35 we correct BANFE-2 line 33, we correct the capital letter.

    1. Not capitalize the first letter for keywords.

    Response: We correct the capital letters.

    1. Psychosis is not a topic of “mental health”. It is a topic of mental disorders or psychiatry. “Mental health” is an anti-psychiatric myth that had terrible consequences in various health systems around the world, especially in the United States of America. The concept should be avoided in current psychiatric literature

    Response: We appreciate the perspective of the reviewer. We agree that the antipsychiatry movements have had negative consequences in health systems around the world, especially in the USA. We also agree that psychosis is a topic of mental disorders, and a topic for psychiatry. However, we do believe that the explanation and response to this severe disorder requires an interdisciplinary effort including psychiatrists, psychologists, nurses, social workers, as well as researchers in several basic and applied fields. We think that the scientific concept of psychopathology is adequate to frame this interdisciplinary effort. These are the changes in the corrected manuscript: The scientific explanation of psychosis remains a challenge with significant implications for clinical practice in the field of psychopathology. (line 48).

    1. Psychosis is not only a problem in the evaluation of delusions and hallucinations. It is also a problem in distinguishing the self from the non-self phenomena. Some authors call this a lack in the grasp of self-properties.

    Response: We appreciate this important comment. We agree that the problem of self-representation and self-recognition is at the core of schizophrenia psychopathology. Thus, we have added two references to the neuroscientific and psychopathological research on disordered self-awareness in the context of schizophrenia (Frith, 2005; Sandsten et al., 2020), and we changed the manuscript, as follows: (line The explanation of the clinical features of psychosis in schizophrenia requires a comprehensive approach not only to symptoms as delusions and hallucinations, but also of the psychopathology of self-awareness in schizophrenia (Frith, 2005; Sandsten et al., 2020). In this article, we will focus on psychotic features as these are measured by the PANSS instrument, with a particular emphasis on delusions as these phenomena have been widely discussed by the tradition of cognitive neuroscience. For instance, in the cognitive framework a two-factor account has been proposed as an explanatory model for the formation and maintenance of delusions in patients across the spectrum of psychosis

    1. All citation regarding “late-onset schizophrenia” soul be re-evaluated with high critics.

    Response: Thanks for this valuable comment. We change de term to late-onset schizophrenia-like. We have included a paragraph: A note of caution should be included for studies of so-called “late onset schizophrenia”, and even more for studies on “very late onset schizophrenia”, as some of the patients included in those samples may suffer from psychosis as a result of neurological disease, instead of primary psychosis due to schizophrenia. Line 84-87

    1. Please, provide some insights explaining the difference between metacognition, insight, anosognosia and theory of mind. The readers will ask themselves if there are any overlapping between these important concepts regarding psychosis. Response: These are helpful suggestions. Thus, we have included a paragraph clarifying these concepts, we add: The term metacognition refers to the processes that enable us to evaluate our own mental operations through monitoring, self-regulation, and control of cognitive mechanisms. This leads to the capacity to recognize accurately, reflect, and regulate on our own mental representations [17]. The metacognitive skills allow the person to identify a particular mental state, for example: experiencing a particular visual or auditory perception, executing a motor act, or accurately experiencing the evocation of information. This has been called metacognitive awareness [1,17,18]. Also, these skills allow us to make judgments about these phenomena and compare incoming information to modify or rectify the judgment. This is called metacognitive confidence [17,19,14,16]. Finally, there is a predictive aspect in metacognition, which enable us to proofread to our own cognitive performance [16]. Metacognitive dysfunction has related with poor outcomes in educational, work and social behavior. At the clinical level, a significant association between metacognitive deficits and the maintenance of delusional beliefs has been shown [20]. For research and practical purposes, it is important to distinguish between metacognition and other related concept, like anosognosia, insight, and theory of mind.  We regard anosognosia as the unawareness of a functional deficit, following focal or diffuse brain damage, (Branch Coslett, 2005; Jenkinson & Fotopoulou, 2014) whereas the term insight refers to a type of self knowledge, “which individuals hold not only about the disorder affecting them but also about how the disorder affects their interaction with the world.” (Marková & Berrios, 1992). Theory of mind, on the other hand, can be definied as the use “of folk psychological knowledge and heuristics to think about one’s own and other people’s mental states”.(Quesque et al., 2024) In this article, we will focus on the concept of metacognition.

    1. Starting a new paragraph when explain the aims of the study.

    Response: We modify the paragraph in line 128-135: This study was focused on the assessment of metacognition in patients with schizophrenia and to describe a new approach to measuring the metamemory phenomenon in a clinical setting with a a word list learning task. The aims of the study were: 1) to measure the overconfidence in metacognitive judgments through the prediction of word list recall; 2) to analyze the correlation between basic neurocognition (memory and executive function), metacognition (through a metamemory test) and the severity of psychotic symptoms. We expect a positive correlation between errors in recall prediction and the severity of the positive symptom factor of the PANSS; also, we expect that overconfidence explains a higher variance in positive symptoms, through a prediction model. 3) Finally, we expected to find a correlation between overconfidence scores and the levels of functionality. And delete de PANNS acronym.

    1. Please delete the reference to the PANSS acronym in the aim of the study.

    Response: we delete PANSS of the aim of the study

    1. Material and methods. Whenever presenting local or national institutions written in Spanish I would recommend writing firs the original designation, in second place to explain de acronym, use italics always for non-English words.

    Response: we modify these points in lines:  139-140.

    1. Patients with more than 65 years old should also be excludes. Below 18-year-olds we deal with child and adolescent psychiatry.

    Response: We appreciate this suggestion. It is documented that there is a relationship between age and metacognitive performance, in that older adults tend to underestimate their overall performance with age. (McWilliams et al., 2023) However, we have only one 67-year-old patient in the sample. This subject was studies with the proper psychiatric and neuropsychological methods to rule out a major neurocognitive disorder, or a neurological disease of any kind (vascular, degenerative, and others). Also, this subject was not a case of so-called “late onset schizophrenia”, but in fact he was a case of schizophrenia followed through many years in the National Institute of Psychiatry. Moreover, the modified manuscript includes a new statistical analysis which shows that age does not correlate with the relevant clinical and neuropsychological variables in our sample. In the new manuscript, have now clarified these points, as follows: We also excluded patients with “late onset schizophrenia”, “very late onset schizophrenia”.

    1. Provide the percentage of patients that were submitted to complete organic differential diagnosis.

    Response: In our Institution, all patients with a psychotic disorder undergo a complete process of differential diagnosis to rule out a neurological disease, including a physical and neurological examination, a structural MRI study and a EEG study; also, the patients in our sample underwent a complete neuropsychological assessment to rule our a major neurocognitive disorder. Regarding lumbar puncture to rule out brain infections or autoimmune encephalitis, the recommendation from experts is that this studies are done in patients with psychosis and with red flags or criteria for autoimmune psychosis (Dalmau et al., 2019; Pollak et al., 2020). Through a partnership with the National Institute of Neurology and Neurosurgery, all patients with red flags are referred to that Institution for further assessment and treatment. The specific system of red flags that is used in this collaboration has been reported previously.(Espinola-Nadurille et al., 2023). None of the subjects included in this study presented with red flags or fulfilled criteria for possible autoimmune psychosis or another neurological disease underlying the psychotic disorder. Dalmau, J., Armangué, T., Planagumà, J., Radosevic, M., Mannara, F., Leypoldt, F., Geis, C., Lancaster, E., Titulaer, M. J., Rosenfeld, M. R., & Graus, F. (2019). An update on anti-NMDA receptor encephalitis for neurologists and psychiatrists: mechanisms and models. The Lancet Neurology. https://doi.org/10.1016/s1474-4422(19)30244-3

    Espinola-Nadurille, M., Restrepo-Martínez, M., Bayliss, L., Flores-Montes, E., Rivas-Alonso, V., Vargas-Cañas, S., Hernández, L., Martínez-Juarez, I., Gonzalez-Aguilar, A., Solis-Vivanco, R., Fricchione, G. L., Flores-Rivera, J., & Ramirez-Bermudez, J. (2023). Neuropsychiatric phenotypes of anti-NMDAR encephalitis: A prospective study. Psychological Medicine, 53(9), 4266–4274. https://doi.org/10.1017/S0033291722001027

    We add: We excluded participants with delusional disorder, substance induced psychosis, schizoaffective disorder, major neurocognitive disorder or any other serious psychiatric comorbidity, catatonic syndrome, or active consumption of any substance other than tobacco. Severe or uncontrolled medical illnesses were also discarded. Also, we used strict criteria to exclude patients with a reasonable suspicion of suffering from an underlying neurological or systemic cause for their psychotic state, including endocrine disease, nutritional pathologies, autoimmune pathologies. We also excluded patients with “late onset schizophrenia”, “very late onset schizophrenia”.

    1. Provide variables such as smoke pack year, body mass index, chlorpromazine equivalents, duration for untreated psychosis, global clinical impression and personal and social performance.

    Response these variables were not taken during the interview with the participants, and therefore were not recorded. We will place this as a limitation of the study.

    1. At line 149 I found a new writing style.

    Response: We correct the text size/type

    1. Explain what kind of statistics was used.

    Response: We add Analysis was performed using the program JASP version 18.0; the significance level was set at 0.05. in line 173

    1. Table 1 is too big. Divide in three smaller tables.

    Response: we divide in two tables, and we place the pharmacological treatment in the text

    1. Explain all acronyms.

    Response: We add. FAST: The Brief Functioning Assessment Scale, line 186

    1. Review your English. Instead of years, it seem that years would be more appropriated.

    Response: We correct that point

    1. Never use the concept illness, sickness, disease. I would recommend use disorder:

    Response: the term illness is used specifically in the name of a test, which we cannot modify, and in the term duration of illness to refer to the number of years with the disorder. This term of chronicity is widely used in the literature in schizophrenia. We chance the term illness duration for disorder duration (table 2) The other terms were not used in the text.

    1. Do not use color in your article, for ecological and aesthetics reason. I would recommend gray scale instead.

    Response: We chance to gray scale in figure 1.

    1. I cannot understand what BADE acronym means.

    Response:  BADE is defined in the introduction section on line 275. bias against disconfirmatory evidence (BADE)

Reviewer 2 Report

Comments and Suggestions for Authors

Thank you for giving me the opportunity to review this manuscript.

1)       Please describe the study design clearly. Was this a cross-sectional study? Please describe the study design by using the PECO (the population, the exposure, the control, and the outcome).

2)       Please describe how the sample size was arrived at. Please describe all predictors, potential confounders, and effect modifiers. I think it is impossible to assess the correlation only by this study design.

3)       Please describe all statistical method including those to control for confounders. Please describe characteristics of study participants on exposures and potential confounders clearly.

4)       Please describe how missing data was handled.

5)       Please describe unadjusted estimates and confounders-adjusted estimates and their precisions if possible.

6)       I think the novelty of this study was quite unclear. Please describe what previous studies found, what they did not find, and what was the novelty of this study.

I think it is necessary to revise the manuscript.

Author Response

We are very grateful to our reviewer, the comments helped us to improve our text and below are the answers to his comments.

  • Please describe the study design clearly. Was this a cross-sectional study? Please describe the study design by using the PECO (the population, the exposure, the control, and the outcome).

Response: Our study design was cross-sectional, which is considered an observational study. According to your suggestion we added a sentence and a reference of this study design. (line 152). In this case the terminology PECO did not apply because our population (patients with schizophrenia) were not exposed to any variable, and we have not a control group. Therefore, we decided to use the classical terminology of observational studies.

  • Please describe how the sample size was arrived at. Please describe all predictors, potential confounders, and effect modifiers. I think it is impossible to assess the correlation only by this study design.

Response: Thank you for your valuable feedback. Convenience sampling was used to recruit participants (line 152) B) We considered predictors as demographic variables and clinical characteristics of our patients. Potential confounders could be the pharmacological treatment, but all our patients were treated and had similarities in the memory assessments, which could indicate that this confounder has no effect in our analysis.  We performed an additional correlation between clinical and demographic variables with the variables of interest, namely psychotic symptoms and overconfidence, to detect possible confounders in analysis. Lines 191-194 and 217-220.

  • Please describe all statistical method including those to control for confounders. Please describe characteristics of study participants on exposures and potential confounders clearly. Response: We added the correlation analysis with demographic and clinical variables such as time of evolution, age and schooling, age of onset and illness duration with overconfidence, psychotic symptoms and total recall to control for possible cofounders (Lines 191-194 and 217-220). We did no find correlation with these possible confounders, so they were not added to the subsequent linear regression analysis.
  • Please describe how missing data was handled.      Response: All participant data were complete (n=51) because the study was designed to ensure the comprehensive evaluation of all included participants. Fortunately, we did not encounter any missing data.
  • Please describe unadjusted estimates and confounders-adjusted estimates and their precisions if possible.

Response: We controlled the regression model, including only those cognitive, metacognitive, clinical or demographic performance variables that will show correlation (prior to Bonferroni correlation) with positive symptoms. When these correlations were performed, only overconfidence performance, total metamemory errors and intrusions were found to be significant.

  • I think the novelty of this study was quite unclear. Please describe what previous studies found, what they did not find, and what was the novelty of this study.

Response. We described in lines 107 to 118 the existing approaches to measure the metacognitive phenomenon in patients with schizophrenia and their difficulties due to the time and resources they require. We added in line 119 that: a novel approach to measure this phenomenon in patients with schizophrenia in clinical settings is the use of a word list.

  • I think it is necessary to revise the manuscript.

Response: Thak you for your comment, by adding the analyses suggested by you and the comments, we consider that the quality of the text has improved.

Reviewer 3 Report

Comments and Suggestions for Authors

The authors of the article attempt to examine the relationship between psychotic symptoms and metacognition in the case of overconfidence in word recall prediction.

The abstract seems a bit long and not well structured. The reader cannot quickly orient himself in the conducted research.

The introduction of the article is very messy. There is no introductory part to acquaint the reader with the basic guidelines in the use of metacognition and cognition and the differences between them. A sufficiently comprehensive review of the literature as well as an analysis of the contradictions in it is lacking.

From this analysis it is necessary to derive the task of the conducted research. It lacks an exact wording of the task.

It is necessary to evaluate cognitive changes with an analysis of the literature on changes in fixation, reproduction and attention in patients with schizophrenia. An evaluation of how these changes would affect the state of metacognition in the case of word recall prediction is needed.

How to consider the term insight used in clinical practice in patients with schizophrenia and the concept of metacognition?

This kind of analysis needs to be done in the introduction.

Materials and methods

Participants. It is necessary to clarify the minimum education of 6 years that is included in the criteria. What does this education of 6 years give. Why did you choose this metric?

The methods used are well described and clearly presented. The statistical methods need further clarification.

In the demographics column, the duration of schizophrenia has a wide range. This also raises the question of varying degrees of cognitive impairment, because there is sufficient evidence that cognitive abilities deteriorate over the course of the disease. This has to be discused.

Results: Results are presented clearly. What is the correlation is between impairments in cognition and metacognition. If no such assessment has been made, this should be presented as a limitation of the study in the Limitations section.

When conducting statistical methods, it is not clear what the variables used are. The visualization of the results is well prepared.

Table 2 needs some more explanations because it is a summary

The discussion is scattered as is the introduction. It is not clear what is being proven and established and how this fits and is analyzed in the context of the data from the literature. No analysis was made of the possible impact of the duration of the disease as well as its onset on metacognitive functions such as confidence in word recall prediction. Are these indicators /duration and the onset/ included in the statistical analysis? How this result can be used in practice?

There is no conclusion from which to derive the message to the reader.

The list of the references has to be increased.

The reviewer

Author Response

We are very grateful to our reviewer, the comments helped us to improve our text and below are the answers to his comments.

The authors of the article attempt to examine the relationship between psychotic symptoms and metacognition in the case of overconfidence in word recall prediction.

  • The abstract seems a bit long and not well structured. The reader cannot quickly orient himself in the conducted research.

Response: Thank you for your comment, we changed the abstract to:

A two-factor account has been proposed as an explanatory model for the formation and maintenance of delusions. The first factor refers to a neurocognitive process leading to a significant change in subjective experience; the second factor is regarded as a failure in hypothesis evaluation characterized by an impairment of metacognitive ability. This study focused on the assessment of metacognition in patients with schizophrenia. The aims of the study were to measure overconfidence in metacognitive judgments through the prediction of word list recall and to analyze the correlation between basic neurocognition (memory and executive function), metacognition through a metamemory test, and the severity of psychotic symptoms.

Method: 51 participants diagnosed with schizophrenia were evaluated. The PANSS was used to assess the severity of psychiatric symptoms, and the Metamemory subtest included in the BANFE Battery was used to evaluate overconfidence, underestimation errors, intrusion and perseverative responses, and total volume of recall.

Results: The strongest correlation observed was between overconfidence errors and the positive factor of the PANSS (r = 0.774, p < 0.001). In the entire model of multiple linear regression (r = 0.78, r² = 0.61; F = 24.57, p < 0.001), the only significant predictor was overconfidence errors.

Conclusion: Our results highlight the relevance of a metacognitive bias of overconfidence, which is strongly correlated with psychotic symptoms and supports the hypothesis that metacognitive defects contribute to the failure to reject contradictory evidence. From our perspective, these findings align with current mechanistic models of schizophrenia that focus on the role of the prefrontal cortex.

  • The introduction of the article is very messy. There is no introductory part to acquaint the reader with the basic guidelines in the use of metacognition and cognition and the differences between them. A sufficiently comprehensive review of the literature as well as an analysis of the contradictions in it is lacking.

Response: We modify the organization of the text, to discuss first the models of the generation of psychosis. Secondly about the neurocognitive factor and thirdly about the metacognitive factor in schizophrenia.

  • From this analysis it is necessary to derive the task of the conducted research. It lacks an exact wording of the task.

Response: a) we describe in lines 107 to 118 the existing approaches to measure the metacognitive phenomenon in patients with schizophrenia and their difficulties due to the time and resources they require. We added in line 119 that: a novel approach to measure this phenomenon in patients with schizophrenia in clinical settings is the use of a word list. B) The description of the task is placed on line 162 to 170, but the precise list of words used has been added in the method section (pear, tube, cow, boat, eraser, sandpaper, hand, bow, letter) line 163-164

  • It is necessary to evaluate cognitive changes with an analysis of the literature on changes in fixation, reproduction and attention in patients with schizophrenia. An evaluation of how these changes would affect the state of metacognition in the case of word recall prediction is needed.

Response: We add in discussion section: Also, the role of basic functions such as attention should be considered into the metacognitive phenomenon; however this topic remains in an intense debate. Matthews and colleagues (48) explore the co-dependency between top-down attention in the performance of visual discrimination task and confidence rating to calculate metacognition. They show that in a dual task, in which individuals must simultaneously attend to two tasks and two types of stimuli, confidence in perceptual judgments remains identical on correct and incorrect trials (presence and absence of attention) but showed a significant correlation between correct trials and attention. Likewise, the authors indicate that the metacognitive accuracies are much higher on tasks of lower complexity and decrease with increasing cognitive demand.  However, Rech, Mamassian and Gardelle (49)  show in their work that confidence judgments are sensitive to involuntary changes in attention; that is, individuals may show greater confidence in their assertions when cues are presented and facilitate the attentional process. To solve this dilemma in our task, we require to perform a specific measure of attention in our patients, this is considered a limitation of the study, but also a hypothesis to explore in this type of measure (line 291-301)

  • How to consider the term insight used in clinical practice in patients with schizophrenia and the concept of metacognition? This kind of analysis needs to be done in the introduction.

Response. We add: Metacognition has also been shown to play a moderating role in other complex phenomena such as clinical insight or the tendency to be unaware of what others perceive as changes or alterations. Lysaker & Chernov [56] showed the first model of overlap between clinical insight, cognitive insight, and metacognition. The results indicate that self-reflection and clinical insight were significantly mediated by metacognition and positive symptoms. An interesting line to develop is the relationship between the measure proposed in the present study and clinical insight. (line 333-337)

Materials and methods

  • It is necessary to clarify the minimum education of 6 years that is included in the criteria. What does this education of 6 years give. Why did you choose this metric? Response In the country of the assessment, basic education includes 6 years, and a person who has completed primary education is capable of adequately responding to an evaluation such as the one conducted in this study. Therefore, the findings would not be attributable to a lack of academic preparation but rather to the cognitive impairments characteristic of the disease.

  • The methods used are well described and clearly presented. The statistical methods need further clarification.                                        Response For a better description we changed the statistical analysis section as follows: Descriptive statistics were performed for clinical and sociodemographic data, using media and proportions as apply.   Pearson correlation test was used to assess the relationships between positive and negative factors of the PANSS and FAST scores with the overconfidence errors, underestimation errors, total recall of memory assessment, the number of perseverative and intrusive responses; we used Bonferrioni for multiple comparisons to correct the correlations. An additional correlation analysis was carried out with the variable’s positive symptoms, overconfidence errors, under-estimations errors and total memory recall with age, years of educations, age of onset and illness duration to identify cofounders. A multiple lineal regression was conducted with variables significantly related to the positive symptoms. Analysis was performed using the program JASP version 18.0; the significance level was set at 0.05.

  • In the demographics column, the duration of schizophrenia has a wide range. This also raises the question of varying degrees of cognitive impairment, because there is sufficient evidence that cognitive abilities deteriorate over the course of the disease. This has to be discused. Response: To resolve this potential confounder, we added additional analysis of illness duration and its influence on overestimation errors and positive symptoms but we did no find significant correlation. On the other hand, although many patients had impairment in the total recall disturbance, this performance also showed no significant correlation with overconfidence or positive symptoms.

  • Results: Results are presented clearly. What is the correlation is between impairments in cognition and metacognition. If no such assessment has been made, this should be presented as a limitation of the study in the Limitations section.                         We add: No significant correlations are observed between memory performance and overestimation (r= -0.34) or underestimation errors either (r=0.26) Line 198-200

  • When conducting statistical methods, it is not clear what the variables used are. The visualization of the results is well prepared.

Response: We add in the statistical methods: Descriptive statistics were performed for clinical and sociodemographic data, using media and proportions as apply.   Pearson correlation test was used to assess the relationships between positive and negative factors of the PANSS and FAST scores with the overconfidence errors, underestimation errors, total recall of memory assessment, the number of perseverative and intrusive responses; we used Bonferrioni for multiple comparisons to correct the correlations. An additional correlation analysis was carried out with the variable’s positive symptoms, overconfidence errors, under-estimations errors and total memory recall with age, years of educations, age of onset and illness duration to identify cofounders. For multiple lineal regression we add: The multiple lineal regression model for predicting positive symptoms was conducted with variables significantly related to the positive symptoms: overconfidence errors (r=0.794***), total errors of metamemory test (r=0.417**) and the intrusion response (r= 0.356**).

  • Table 2 needs some more explanations because it is a summary

Response: We add in table 3, line: The model was constructed using positive symptoms as dependent variable, and overconfidence errors, total or metacognitive errors and intrusion response as predictors. We observed that the model explains 78% of the variance, being only the positive errors a significant predictor of the positive symptoms of the PANSS

  • The discussion is scattered as is the introduction. It is not clear what is being proven and established and how this fits and is analyzed in the context of the data from the literature. No analysis was made of the possible impact of the duration of the disease as well as its onset on metacognitive functions such as confidence in word recall prediction. Are these indicators /duration and the onset/ included in the statistical analysis? How this result can be used in practice?

Response: a) we broaden the discussion in an attempt to more accurately contextualize our findings. b) We add:  The correlations between PANNS positive scores and overconfidence errors with sociodemographics and clinical variables age (r=0.18, r=0.06 ), years of education (r= -0.08, r= -0.10), age of onset (r= -0.07, r= -0.08) or illnes durations (r=0.21, r=0.10) were no significants. No significant correlations are observed between memory performance and overestimation (r= -0.34) or underestimation errors either (r=0.26), line 190-194

  • There is no conclusion from which to derive the message to the reader

Response. We add: In conclusion, overconfidence, understood as a tendency to have excessive confidence in one's own beliefs and judgments, is related to the presence of positive symptoms in patients with schizophrenia. This finding suggests that in the treatment of patients with schizophrenia, evaluating this dimension of metacognition could allow us to explore other interventions that help reduce positive symptoms.

  • The list of the references has to be increased.

Response: We add:

Matthews J, Schröder P, Kaunitz L, van Boxtel JJA, Tsuchiya N. Conscious access in the near absence of attention: critical extensions on the dual-task paradigm. Philos Trans R Soc Lond B Biol Sci. 2018;373(1755):20170352. doi:10.1098/rstb.2017.0352

Recht S, Mamassian P, de Gardelle V. Metacognition tracks sensitivity following involuntary shifts of visual attention. Psychon Bull Rev. 2023;30(3):1136-1147. doi:10.3758/s13423-022-02212-y 50           

Lysaker PH, Chernov N, Moiseeva T, et al. Clinical insight, cognitive insight and metacogni-tion in psychosis: Evidence of mediation. J Psychiatr Res. 2021;140:1-6. doi:10.1016/j.jpsychires.2021.05.030

Frith, C. The self in action: Lessons from delusions of control. Consciousness and Cognition, 2005; 14(4), 752–770. https://doi.org/10.1016/j.concog.2005.04.002.

Sandsten, K. E., Nordgaard, J., Kjaer, T. W., Gallese, V., Ardizzi, M., Ferroni, F., Petersen, J., & Parnas, J. Altered self-recognition in patients with schizophrenia. Schizophrenia Research. 2020; 218, 116–123. doi: 10.1016/j.schres2020.01.022

Branch Coslett, H. Anosognosia and body representations forty years later. Cortex. 2005; 41(2), 263–270. doi.org/10.1016/s0010-9452(08)70912-2

Jenkinson, P. M., & Fotopoulou, A. Understanding Babinski’s anosognosia: 100 years later. Cortex. 2014; 61, pp. 1–4. Masson SpA. doi.org/10.1016/j.cortex.2014.10.005

Marková, I. S., & Berrios, G. E. The assessment of insight in clinical psychiatry: a new scale. Acta Psychiatrica Scandinavica.; 1992 86(2), 159–164. doi.org/10.1111/j.1600-0447.1992.tb03245.x

Quesque, F., Apperly, I., Baillargeon, R., Baron-Cohen, S., Becchio, C., Bekkering, H., Bernstein, D., Bertoux, M., Bird, G., Bukowski, H., Burgmer, P., Carruthers, P., Catmur, C., Dziobek, I., Epley, N., Erle, T. M., Frith, C., Frith, U., Galang, C. M.,Brass, M. Defining key concepts for mental state attribution. Communications Psychology. 2024; 2(1). doi:10.1038/s44271-024-0077-6

Round 2

Reviewer 2 Report

Comments and Suggestions for Authors

I think this manuscript would be suitable for publication in this journal.

Author Response

We appreciate your help to improve our text.

Reviewer 3 Report

Comments and Suggestions for Authors The authors have taken into account the critical remarks of the reviewer. Submitted in this way, the article can be published. There is a spelling error on line 342 the second word. Successful day. The reviewer

Author Response

(The authors gave the same response as above.)
